# How Does the Historic Built Environment Influence Residents' Satisfaction? Using Gradient Boosting Decision Trees to Identify Critical Factors and the Threshold Effects

**Xian Ji** [1], **Yu Du** [1,*] and **Qi Li** [2,*]

1   Jangho Architecture College, Northeastern University, Shenyang 110169, China; jix@mail.neu.edu.cn
2   School of Architecture, Harbin Institute of Technology, Shenzhen 518055, China
*   Correspondence: duy@mail.neu.edu.cn (Y.D.); li7_phdhit@126.com (Q.L.)

**Abstract:** Historic cities, rich in heritage values and evocative of collective memories and meanings, also constitute crucial living environments for urban residents. These cities increasingly face challenges from urbanization and globalization, leading to cultural discontinuities and the homogenization of cityscapes. Such developments can diminish residents' sense of belonging and identification with their cities. Emphasizing local residents' satisfaction is thus essential to urban conservation. This study, using data from Dandong, China, employs Gradient Boosting Decision Trees (GBDT) to identify factors affecting residents' satisfaction in historic built environments. The analysis reveals that over half of the variability in satisfaction is linked to distinct features of the historic environment. Among the fourteen key influencers identified, contextual order emerges as the most impactful factor, notable for its significant effects and interactions with other variables. This study also uncovers pronounced non-linear effects and thresholds for physically measured characteristics. For instance, open space markedly boosts satisfaction when exceeding 34%, satisfaction diminishes with travel times to heritage sites longer than 6.7 min, and satisfaction decreases when the entropy index for diversity surpasses 0.758. These findings provide critical insights for guiding urban conservation strategies and promoting a data-driven approach to enhance residents' satisfaction in historic urban settings.

**Keywords:** urban heritage conservation; historic built environment; satisfaction; gradient boosting decision trees; nonlinear association; threshold effect

## 1. Introduction

Historic cities are embodiments of historical layering, where natural and cultural values and attributes have been interwoven and accumulated over time by generations, adapting to ever-changing contexts [1]. Historic cities, rich in unique historical heritages, feature landscapes of special value and significance to the cultural groups residing within them. These historic urban landscapes are subject to constant transformation driven by dynamic forces in response to evolving urban development needs [2]. These landscapes form the core living environment for urban residents. However, the forces of urbanization and globalization have posed significant challenges to historic cities, leading to issues like cultural discontinuities and the homogenization of cityscapes, thereby eroding residents' sense of belonging and identification with their cities [3]. Additionally, the imbalance between heritage preservation and urban development has adversely affected the quality of life at heritage sites [4,5]. Urban heritage, bridging history and the present, is crucial to local identity. Both tangible and intangible urban heritages, along with their associated memories, showcase the ongoing shaping and reshaping of urban landscapes [6]. These heritages, as potential catalysts for social cohesion and urban regeneration, hold immense value [7].

Conservation efforts in built environments have frequently fallen short of success and sustainability [8]. A fundamental issue underpinning these challenges is the absence of

a unified consensus regarding the objectives of conservation—specifically; what should be preserved; the underlying rationale for its preservation; and the methodologies to be employed in its conservation [9].

The field of urban conservation, gaining momentum post-French Revolution, has seen a paradigm shift in its approach, evolving from a focus on physical objects and visual aspects to emphasizing rituals, experiences, and empathy [10]. This transition reflects a growing trend of viewing the human environment as a cultural landscape, managed within landscape discourse [11]. This holistic view aligns urban conservation with urban development, with the joint goal of enhancing human environments and fostering community prosperity, thus positioning urban conservation as a vital component of urban sustainability [12].

The psychological motivation behind heritage conservation is rooted in a desire for security, encompassing both loss prevention and assurance of continuity [13]. Early heritage conservation efforts perceived change as a threat, destabilizing this sense of security. However, with the introduction of the Historic Urban Landscape (HUL) concept and the subsequent Recommendation on the Historic Urban Landscape, managing change has become a primary tool in the conservation of historic urban areas [14]. This shift signifies a more dynamic and adaptive approach to heritage conservation, one that embraces and manages changes rather than opposing them.

Navigating the complex balance between preservation and development in historic urban environments involves a dynamic interplay between change and constancy. Embracing change is essential, but it is equally crucial to ensure that any alterations to historical settings are carried out with restraint and appropriateness. The needs and expectations of local residents, who are both integral to and active participants in shaping their environment, should be a priority. Involving the community in decision-making processes and focusing on their satisfaction with historic urban settings are pivotal in guiding urban conservation and regeneration initiatives.

For making informed decisions about conserving and changing the historic built environment, it is necessary to understand how it impacts the people living within it. However, there is a dearth of studies exploring the relationship between the historic built environment and residents' satisfaction, with most research focusing only on subjective dimensions [3]. Urban planners need insights not only into the factors most critical to satisfaction but also into the optimal parameters for the physical environment. This knowledge enables the formulation of relevant policies and quantitative guidelines.

Moreover, theories suggest that characteristics of the built environment might have non-linear effects on life satisfaction [15,16]. A non-linear effect implies that the incremental impact of a characteristic on satisfaction varies depending on the characteristic's value. Often, the influence of built environment characteristics on life satisfaction occurs only beyond a certain threshold, not across the entire spectrum of these characteristics [17,18]. Understanding this non-linearity and identifying effective ranges can provide planners with insights on how to allocate resources more efficiently. Yet, empirical studies examining the non-linear associations between historic built environment characteristics and residents' satisfaction are scarce, highlighting a gap in current research.

Using the data from the old town of Dandong, China, this research attempts to fill the gaps. The Gradient Boosting Decision Trees (GBDT) technique is employed to explore the factors influencing residents' satisfaction with historic built environments and identify the effective ranges in which key factors correlate with satisfaction. It seeks to answer several key research questions: (1) To what extent does the built environment of historic cities contribute to residents' satisfaction? (2) Which built-environment characteristics play an essential role in generating satisfaction? (3) Do these characteristics have nonlinear or threshold associations with residents' satisfaction? (4) Is there any interaction effect among the built environment characteristics in influencing satisfaction?

This study significantly enriches the existing literature in three key aspects. Firstly, it is among the few to investigate the relationship between the characteristics of the built en-

vironment and the satisfaction of residents in historic settings. Our findings offer valuable insights for managing historic cities, particularly in balancing heritage preservation with the enhancement of residents' place attachment and sense of belonging. Secondly, this research challenges the prevalent linear perspective in the built environment and life satisfaction literature by examining the relative influences of historic built environment attributes and their nonlinear associations with residents' satisfaction. This approach advances the theoretical understanding of how environmental factors correlate with life satisfaction. Thirdly, this study employs a novel combination of the SHAP (SHapley Additive exPlanations) method and Individual Conditional Expectation (ICE) plots. This methodology enhances the interpretability of the GBDT model, illuminating feature variability and interactions. As a result, it provides more comprehensive and nuanced insights for urban planning priorities in historic cities, surpassing previous research in this domain.

In the subsequent section of this paper, we introduce factors related to the historic built environment that influence residents' satisfaction. This is followed by a detailed description of this study area, the survey methodology, the data collection procedures, and the data analysis techniques employed. Subsequently, the implementation of the GBDT model is described, along with a discussion of its performance, including the nonlinear associations and threshold effects it reveals. The interaction effects among variables are then elaborated upon in the final subsection of the Results section. The Discussion section addresses this study's limitations and explores the policy implications of the findings. The concluding part of this paper summarizes this research and underscores the key insights gleaned from this study.

## 2. Materials and Methods

### 2.1. Factors Related to the Historic Built Environment That Affect Residents' Satisfaction

Understanding the intricate relationship between residents and their environment, especially in the context of the historic built environment, is a complex yet vital task. This relationship is dynamic [19], where the level of harmony between urban residents and their daily surroundings plays a crucial role in shaping their satisfaction [20,21]. Satisfaction is a result of how individuals psychologically and physiologically respond to external factors in their environment [20].

The concept of satisfaction, whether related to life as a whole or specifically to the environment, is influenced by the balance between expectations and reality [22]. This balance is key to forming one's satisfaction judgments. Following Campbell's model [23], satisfaction in various aspects of life, including satisfaction with the historic built environment, collectively influences overall life satisfaction. Thus, residents' satisfaction with their historic surroundings is an integral part of their overall well-being. The effectiveness of an environment is measured by how well it resonates with its inhabitants [24], emphasizing the importance of considering residents' perceptions and emotions towards their living spaces in urban planning [25]. Therefore, understanding and prioritizing residents' satisfaction in the historic built environment is not only about managing physical spaces but also about enhancing the overall quality of life of urban residents [3,26,27].

The built environment can be regarded as a service provided to its inhabitants by governments and developers, akin to customer satisfaction in the hospitality industry [28]. Overall satisfaction is derived from an evaluation of various attributes that constitute the subject of this study [29]. Considering the impracticality and unnecessary focus on all attributes to reflect a concept's true complexity, identifying key factors influencing residents' satisfaction with the historic built environment becomes imperative.

Numerous studies have investigated the relationship between environmental characteristics and residential satisfaction, an essential aspect of life satisfaction. Influential factors include personal and social elements, as well as physical attributes [30,31]. For example, Hur, Nasar, and Chun identified building density and vegetation rate as significant factors [32]. Cao highlighted elements like density, diversity, design, and environmental amenities in residential areas [33]. Kaplan underscored the positive impact of natural

elements on neighborhood satisfaction [34], which has also been proven by many other scholars [35,36].

Lynch's identification of five key city elements [37]—paths; edges; districts; nodes; and landmarks—and his principles of vitality [38]; sense; fit; access; and control have been influential in urban studies. Building on Lynch's work, Smith, Nelischer, and Perkins outlined urban community principles such as character, connection, mobility, and diversity [39]. In the field of urban conservation, focus areas include historic urban patterns, heritage preservation, visual linkages, contextual harmony, and the adaptive reuse of cultural heritage from an aesthetic perspective [40–44]. Themes like heritage accessibility, utilization types, interpretation facilities, and the impact of tourism development are prevalent in studies on heritage value and revitalization [45–48].

The relationship between historic built environments and wellbeing is inherently interdisciplinary, yet there remains a notable scarcity of research specifically addressing subjective wellbeing in this context [49]. Given the absence of a specific measurement scale for the impact of the historic built environment on residents' subjective perceptions, this study developed its related dimensions and attributes by drawing insights from built environment satisfaction, residential satisfaction, urban planning principles, and urban conservation and revitalization. A focus group comprising three experts in heritage preservation and urban planning was formed to develop and validate the measurement scale. Consequently, fifteen attributes across four dimensions were selected for further research, as illustrated in Figure 1 and Table 1.

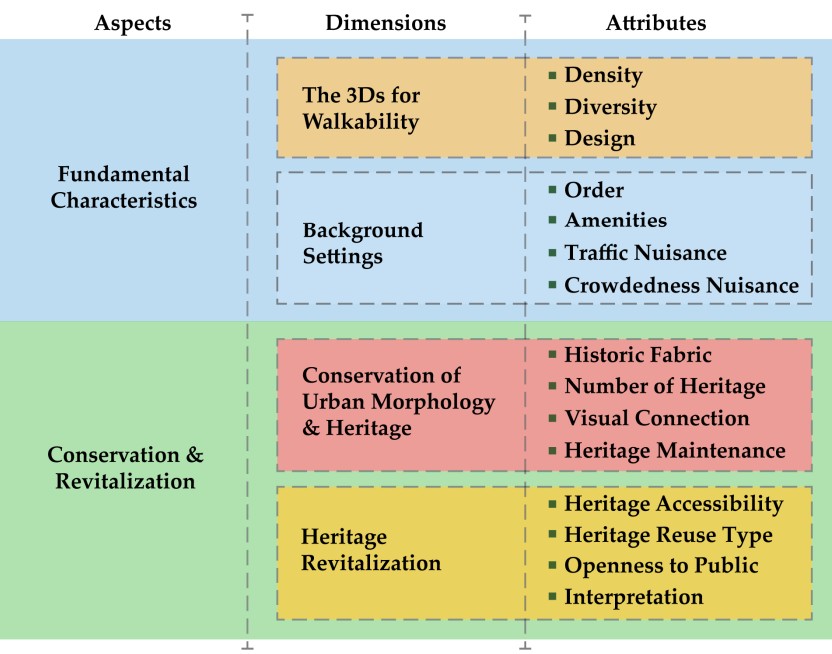

**Figure 1.** Historic built-environmental attributes were selected by the focus group.

### 2.2. This Study Area and Data Collection

Dandong, China's largest border city, is situated along the Yalu River, marking the boundary between China and North Korea. Its significant location, providing easy access to the sea and the Korean Peninsula, has played a pivotal role in its extensive and influential history. This city's evolution, shaped by its distinctive geographical setting and the regular interaction of different cultures and ethnicities, has given rise to its unique Historic Urban Landscape. Numerous urban heritage sites have emerged as iconic landmarks in Dandong, with the Yalu River Broken Bridge and the easternmost point of the Great Wall being the most renowned.

**Table 1.** Description of the historic built-environmental attributes selected by the focus group.

| Attributes | Description |
| --- | --- |
| Density | Density is measured as building density, reflecting the concentration of built environments, calculated by dividing building square footage by land area. |
| Diversity | Diversity refers to the variety and mix of different uses and functions within an urban area, quantified using the entropy index based on POI data from Baidu Maps. |
| Design | This metric, measured by Block Density, reflects urban physical design impacting street accessibility and connectivity. |
| Order | Order embodies the essence of historic built environments, encompassing aesthetics, contextual harmony, and balance between old and new, reflecting the overall atmosphere and character of historic urban areas. |
| Amenities | The proportion of open space represents environmental Amenities. |
| Traffic Nuisance | Measures the level of disruption caused by vehicular traffic, including noise, congestion, and air pollution, impacting the urban living experience. |
| Crowdedness Nuisance | Assesses the extent of discomfort due to high pedestrian density, reflecting on the impact of overpopulation and limited space in urban areas. |
| Historic Fabric | Represents the preservation of historic urban structure or urban fabric, quantified by the percentage of preserved historical road network patterns. |
| Number of Heritage | Quantifies the number of heritage sites or elements within an area, indicating the presence of historical and cultural landmarks. |
| Visual Connection | Measured by the proportion of the area from which heritage sites are visible, indicating visual accessibility to cultural landmarks. |
| Heritage Maintenance | Assesses the condition and upkeep of heritage sites, reflecting efforts to preserve historical integrity. |
| Heritage Accessibility | Calculated based on the average time required to reach heritage sites from residences, indicating ease of access to historical locations. |
| Heritage Reuse Type | Categorizes heritage sites based on their current use, such as commercial, administrative, religious, tourism, or leisure purposes. |
| Openness to Public | Assesses heritage site accessibility, ranging from completely open and free to not open to the public, indicating the level of public engagement allowed. |
| Interpretation | Evaluates the adequacy of interpretive facilities at heritage sites, determining the effectiveness of conveying their historical significance. |

The data for this research came from the built environment and well-being study. We administered a survey to residents living in historic environments in the old town of Dandong in May–July 2023. Five of the most representative historic urban areas were selected as survey locations (Figure 2), each with its own distinct characteristics and a fair amount of heritage.

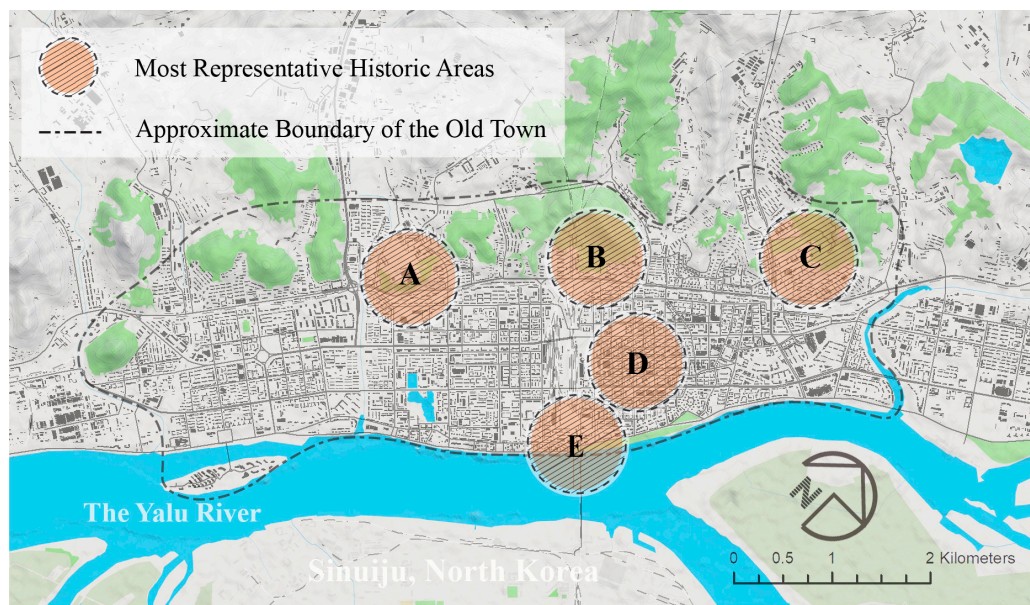

**Figure 2.** Spatial distribution of the selected locations in the old town of Dandong. A: Historic area around the Korean War Memorial; B: Historic area near Jinjiang Mountain Park; C: Historic area around Yuhuang Mountain; D: Historic area near Train Station Square; E: Historic area around the Yalu River Broken Bridge.

The questionnaire was created using a digital tool provided by wjx.cn. Prior to its widespread distribution, it underwent a preliminary test to identify any biases, with participants including students, staff from our School, and acquaintances of this research team. Adjustments to the questionnaire's format and content were made based on this initial feedback. Residents were invited to partake in the survey through distributed leaflets, each containing a cover letter detailing this study's purpose and a QR code. This code enabled quick access to the survey via common smartphones. To ensure a random sample, eight experienced postgraduate students from our team actively sought respondents in diverse locations within the selected areas, including public spaces, shopping districts, residential buildings, and transportation facilities. We conducted a power analysis and found that the sample size should be at least 377. We increased it to 600 to accommodate unexpected data issues. Out of the 600 leaflets distributed, 481 residents responded to the survey, yielding 433 valid, completed questionnaires.

Table 2 presents a comparison of certain demographic characteristics of our sample with those reported in the 2020 census. In our sample, Females and the age group of 18–35 are overrepresented. Conversely, Young people under 18 and the elderly over 60 are underrepresented. Additionally, the Manchu ethnic group is underrepresented, although the percentage distribution of different ethnic groups aligns closely with census data. A notable discrepancy is observed in the educational level of respondents, which is significantly higher compared to the 2020 census figures. Despite these differences, the diversity across various categories within the sample is sufficiently broad, ensuring that the effects on satisfaction are adequately captured. This mitigates concerns about the differences from the census data [33]. While the sample may not perfectly mirror the univariate distributions of the general population, the relationships among the variables studied are robust and can be considered generalizable [50].

**Table 2.** Sample profile and the comparison with the 2020 census.

| Items | Categories | Percentage |
|---|---|---|
| Gender | Female | 63.66% (51.18%) |
| | Male | 36.34% (48.82%) |
| Age | Under 18 | 0.68% (14.61%) |
| | 18–35 | 52.14% (16.55%) |
| | 35–60 | 43.34% (42.34%) |
| | Over 60 | 3.84% (26.50%) |
| Type of Residence | Native-born | 75.17% (N/A) |
| | Migrant | 24.83% (N/A) |
| Length of Residence | <5 years | 4.74% (N/A) |
| | 5–10 years | 5.19% (N/A) |
| | 11–30 years | 39.05% (N/A) |
| | >30 years | 51.02% (N/A) |
| Ethnic Groups | Han | 83.07% (64.10%) |
| | Manchu | 14.67% (31.60%) |
| | Other | 2.26% (4.30%) |
| Education | Middle school and below | 10.16% (61.33%) |
| | High school | 15.80% (18.76%) |
| | Junior college | 25.06% (10.77%) |
| | College | 42.21% (8.53%) |
| | Graduate school | 6.77% (0.61%) |
| Income | 2000 or less | 6.77% (N/A) |
| | 2001–4000 | 22.58% (N/A) |
| | 4001–6000 | 24.83% (N/A) |
| | 6001–10,000 | 29.57% (N/A) |
| | Over 10,000 | 16.25% (N/A) |

Note: The percentage data in parentheses came from the 2020 census data provided by Liaoning Provincial Bureau of Statistics (https://tjj.ln.gov.cn/tjj/tjxx/pcsj/people/pczl/indexch.htm, accessed on 15 November 2023); N/A means this data were not provided by the 2020 census; income here is household monthly income in RMB Yuan.

The questionnaire incorporated four distinct categories of variables: information on the spatial range of daily activities, perceived built-environmental characteristics, overall satisfaction with the built environment, and demographics. For a detailed view, please refer to Appendix B. To assess perceptions of the historic built environment, respondents rated a series of attributes on a seven-point ordinal scale ranging from "extremely not true" (1) to "entirely true" (7). Overall satisfaction was evaluated using a single question, where participants expressed their level of satisfaction with their historic urban environment on a scale from "strongly dissatisfied" (1) to "strongly satisfied" (7). Demographic data collected included age, gender, educational background, income level, duration and type of residence, and ethnic affiliation.

Residents' perceptions of the urban environment show notable heterogeneity. Individuals vary in their familiarity with different urban areas, often showing preferences for certain sectors. This familiarity significantly influences their overall satisfaction with the urban environment. Prior research suggests that the spatial boundaries indicative of a person's place attachment can be delineated based on the spatial range of their daily activities [51]. Thus, we included questions about the spatial range of respondents' daily activities to gather data on locations meeting both their material and immaterial needs, encompassing residences, workplaces, shopping, entertainment, and other relevant places. The place of residence is considered a mandatory response, along with at least three additional bounding points of significance for each respondent.

Subsequently, using data on respondents' daily activity ranges, we employed ArcGIS 10.7 to generate Minimum Convex Polygons (MCPs) that represent their place attachment. These MCPs included each participant's home and at least three additional significant points. We then estimated the objective values of the selected built-environmental attributes within each

respondent's MCP area (Table 3). The indicator of Density used here is the building density. Diversity is represented by the entropy index [52], calculated using POI data from Baidu Maps. Block density is used to measure Design, which impacts the accessibility and connectivity of streets and contributes to urban form and walkability [53,54]. The data for the Number of Heritage and Reuse Types are derived from survey data collected in previous studies and data provided by the government, with Reuse Types represented by the main methods of heritage reuse within the area. Heritage accessibility is calculated based on the average time cost of reaching each heritage within the area from the place of residence. The share of open space is used to represent environmental Amenities. The proportion of open space represents environmental Amenities. The percentage of preserved historical road network patterns serves as the objective metric for Historic Fabric. Visual Connection with Heritage is quantified by the proportion of the area from which heritage sites or historic landmarks are visible.

**Table 3.** Descriptive statistics of the built environment variables.

| Continuous Variable | Unit | Minimum | Maximum | Average | Standard Deviation |
|---|---|---|---|---|---|
| Density | Proportion | 0.11 | 0.23 | 0.16 | 0.03 |
| Diversity | Bits | 0.66 | 0.79 | 0.75 | 0.03 |
| Design | Quantity/km$^2$ | 6.93 | 27.20 | 13.71 | 4.67 |
| Amenities | Proportion | 0.10 | 0.53 | 0.37 | 0.09 |
| Historic Fabric | Proportion | 0.26 | 0.85 | 0.54 | 0.19 |
| Heritage Visibility | Proportion | 0.13 | 0.60 | 0.47 | 0.09 |
| Heritage Accessibility | Minutes | 5.51 | 9.90 | 6.85 | 1.37 |

| Discrete Variable | Unit | Minimum | Maximum | Average | Standard Deviation |
|---|---|---|---|---|---|
| Number of Heritage | Quantity | 2 | 15 | 6.78 | 3.04 |
| Order | N/A | 1 | 7 | 5.27 | 1.65 |
| Crowdedness Nuisance | N/A | 1 | 7 | 5.70 | 1.17 |
| Traffic Nuisance | N/A | 1 | 7 | 5.72 | 1.18 |
| Heritage Maintenance | N/A | 1 | 7 | 5.77 | 1.30 |
| Openness to Public | N/A | 1 | 7 | 5.32 | 1.92 |
| Interpretation | N/A | 1 | 7 | 5.44 | 1.53 |

| Categorical Variable | | | | Categories | Number of Instances |
|---|---|---|---|---|---|
| Reuse Type of Heritage | | | | Tourism and Leisure | 369 |
| | | | | Administration | 23 |
| | | | | Religion | 51 |

Note: For certain discrete variables, the values represent perceived levels of built-environmental characteristics rather than actual measurements. Consequently, the units for these variables are not applicable (N/A).

Heritage Maintenance, Openness to the Public, Interpretation, Order, Traffic Nuisance, and Crowdness Nuisance is also essential in characterizing the historic built environment. Quantifying and measuring these factors, however, is challenging. Thus, this study uses the perception data of these six factors obtained through questionnaires as a representation of the actual physical environmental conditions. Although these are not actual physical parameters, to a certain extent they can reflect the level of the built-environmental characteristics.

It should be noted that, despite our sample originating from five historic urban areas in Dandong's old town, spatial dependency does not pose an issue in this study. Firstly, individual-based measures were employed to capture variables of the built environment. Specifically, we delineated distinct MCP (Minimum Convex Polygon) areas based on the daily activity ranges of respondents and estimated the objective values for these areas. Consequently, respondents residing in the same neighborhoods do not have identical measures.

### 2.3. Analysis Method

This study employs Gradient Boosting Decision Trees to examine the relationships between characteristics of the built environment and residents' satisfaction with the historic urban environment. Originating from the field of computer science [55], the GBDT method is recognized for its robust predictive abilities and its capacity to identify non-linear relationships among variables. Given these advantages, GBDT has become increasingly popular in urban studies and satisfaction research [16,18,56], especially for investigations concerning the built environment [57–59].

Although many machine learning algorithms can handle non-linear and threshold effects, GBDT stands out for its advantages. GBDT often surpasses algorithms like Random Forests in accuracy and overfitting prevention. Unlike Logistic or Linear Regression, it effectively models complex, non-linear relationships without extensive feature engineering. More scalable than Support Vector Machines (SVMs) and offering greater interpretability compared to neural networks, GBDT is also more efficient in handling large, complex datasets than K-Nearest Neighbors (KNN), making it a versatile and effective choice for diverse analytical tasks.

The GBDT approach integrates decision trees with gradient boosting. This offers a robust method for modeling complex nonlinear associations without assuming predefined relationships among variables. The foundation of a GBDT model involves constructing decision trees. GBDT classifies observations using decision trees at various split points, employing the mean response within a leaf for prediction. Figure 3 illustrates a single decision tree that addresses a continuous variable Y and incorporates two predictors, $x_1$ and $x_2$. Initially, the predictive space is divided into a pair of regions, within which the response is estimated by calculating the average of Y for each segment. The selection of the predictor and the division point are optimized for the most accurate fit. Subsequently, either one or both of these initial regions are further bifurcated, with the process persisting until a predetermined termination criterion is met. In the depicted example, the predictive space is segmented into four distinct regions—denoted as R1; R2; R3; and R4—through the application of three division points; labeled $c_1$, $c_2$, and $c_3$. The decision tree model in question generates a prediction for the response Y by assigning a fixed value $c_m$ to each region $R_m$. This process is encapsulated in Equation (1).

$$F_m(x) = \sum_{m=1}^{4} c_m I\{(x_1, x_2) \in R_m\}, \tag{1}$$

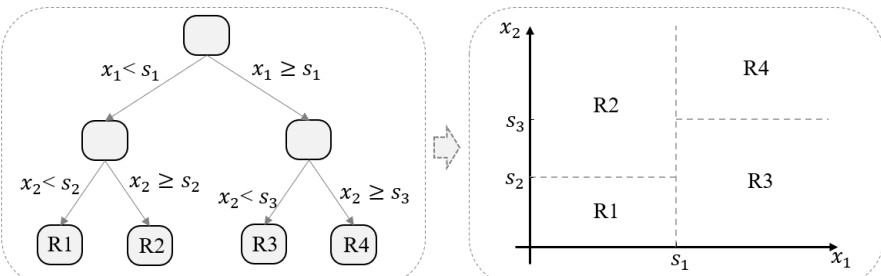

**Figure 3.** An example of the decision tree.

The GBDT model is iteratively built, focusing on minimizing prediction errors through sequential model enhancements. Each tree is developed based on the residuals of the previous tree, thus progressively improving prediction accuracy. Initially, the GBDT model starts with a constant value, typically the mean of the dependent variable, for prediction. Residuals are then computed as the differences between observed and predicted values. A tree is subsequently added to predict these residuals. The revised predicted values of the dependent variable are calculated as the sum of its previous predicted values and the predicted residuals, adjusted by a learning rate. New residuals are determined by subtracting these updated predicted values from the observed values. This iterative process

of adding trees continues until further additions yield no significant improvement in prediction or the maximum pre-set number of trees is reached. Please refer to Appendix A for the mathematical notations of the GBDT algorithm. Furthermore, for an in-depth understanding, please refer to the videos provided in the Supplementary Materials section.

Optimizing the GBDT model requires fine-tuning several key parameters. The maximum tree depth is crucial to managing model complexity and avoiding overfitting. The learning rate determines each tree's impact on the model, with a lower rate improving generalization but requiring more trees. The optimal number of trees needs to be determined to strike a balance between complexity and overfitting. The model's structure is also shaped by setting the minimum samples for leaf nodes and internal node splits, influencing the model's smoothness and ability to capture variable interactions.

Compared to traditional regression models, the GBDT model offers several advantages. The GBDT method, imposing fewer constraints, yields more precise estimates than those obtained through linear regression [17]. It is capable of handling issues of multicollinearity and can accommodate missing values and outliers. More significantly, the ensemble-based boosting approach is effective even with samples of relatively small size [59–62]. The GBDT model does not assume any pre-specified relationship, allowing it to capture complex nonlinear associations, especially when such associations vary among independent variables [18,63]. GBDT models are also capable of generating partial dependence plots (PDPs), which visualize the marginal effect of one or two features on the predicted outcome, independent of other features' values. PDPs are invaluable for interpreting non-linear relationships and interactions between variables.

Unlike traditional regression methods, GBDT does not provide *p*-values and thus does not assess the statistical significance of observed effects, which can be considered a limitation when evaluating the influence of independent variables. However, in the context of non-linear relationships, a linear assumption may lead to flawed conclusions, rendering the *p*-values calculated under incorrect model specifications meaningless and potentially misleading. Moreover, GBDT sheds light on the relative importance of independent variables, which is critical for planning and decision-making. The focus on practical significance often outweighs statistical significance since the real-world impact of a variable is measured by the magnitude of its effect, not merely its statistical detection. Especially in large samples, even a minor effect can achieve statistical significance, making the understanding of the actual influence of variables even more critical [63,64].

## 3. Results

In this study, we utilized the scikit-learn package (version 1.3.2) within the Python 3.10 environment to estimate the GBDT model. The Jupyter Notebook interface (version 6.4.12) from Anaconda was employed to enable an interactive and iterative approach to model building and evaluation.

### 3.1. Model Performance

After dividing the data into dependent and independent variables and performing one-hot encoding for categorical variables, a preliminary model was constructed. Subsequently, we utilized the GridSearchCV algorithm, which combines a comprehensive grid search approach with a five-fold Cross-Validation (CV) procedure, to fine-tune the hyperparameters. The "n_estimators" parameter of the GradientBoostingRegressor class in scikit-learn, representing the number of trees in the forest, was varied in a range from 100 to 400 at increments of 100. The "max_depth" parameter, specifying the maximum depth of each tree, was tested with values doubling from 2 to 10. For "min_samples_split", determining the minimum number of samples required to split an internal node, values from 2 to 5 were explored. The "learning_rate" was examined across a diverse set of values: 0.001, 0.005, 0.01, 0.05, 0.1, 0.5, and 1. In terms of "min_samples_leaf", which sets the minimum number of samples at a leaf node, our exploration included 1, 2, 4, 8, 16, and 20. A consistent random-state of 42 was maintained for reproducibility. Due to computational

constraints preventing the coverage of all possible parameter values, we supplemented GridSearchCV with the BayesSearchCV algorithm. This stochastic optimization algorithm employs Bayesian optimization to identify optimal hyperparameters over a broader range.

The optimal hyperparameter set identified via GridSearchCV included a learning_rate of 0.05, max_depth at 2, min_samples_split of 2, n_estimators at 100, and min_samples_leaf at 16 (the explanations of these parameters are provided in Section 2.3). For further in-depth information, please refer to the document available in the Supplementary Materials section. Ten rounds of BayesSearchCV were conducted, yielding ten sets of optimal parameters. However, the models constructed with these parameters did not outperform the model developed using the optimal hyperparameters from GridSearchCV.

As depicted in Figure 4, the deviance plot demonstrates strong model performance. The training set deviance decreases sharply, indicating rapid initial fit improvement, while the test set deviance reflects a similar trend at a more conservative rate, suggesting effective generalization to new data. Both training and test deviances plateau as iterations increase, showing no signs of overfitting, as evidenced by the stable test deviance. This indicates a well-fitted model for both the training and unseen data, capturing underlying patterns without being tailored to noise or specific data artifacts.

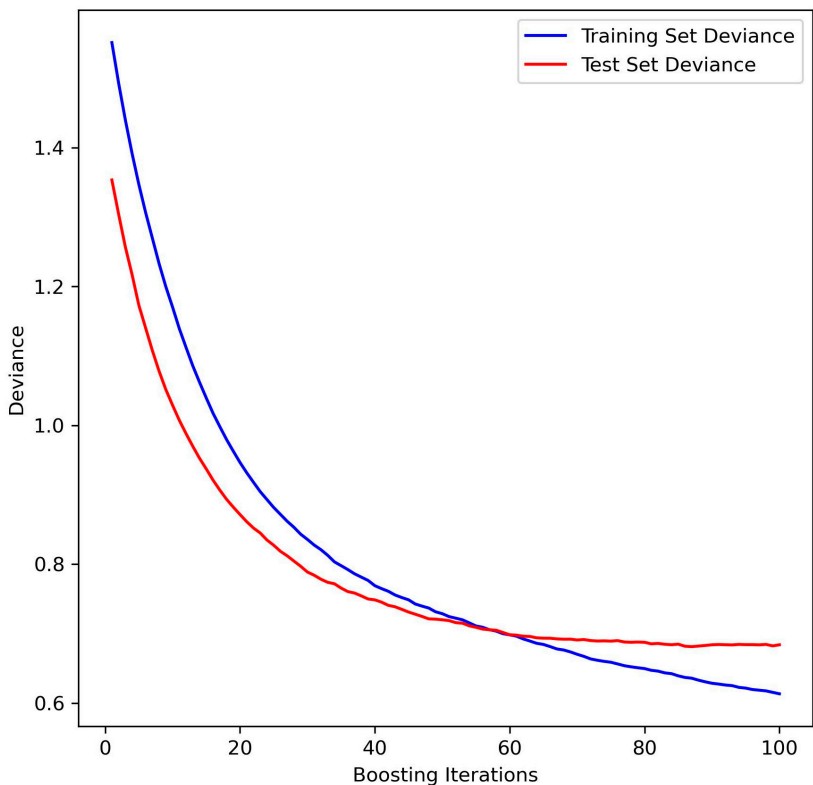

**Figure 4.** The deviance plot of the GBDT model.

Despite the inherent complexities and potential noise within the data, our model achieved an $R^2$ value of 0.5165, and the corresponding Root Mean Squared Error (RMSE) on the test set is 0.8270. This result is particularly noteworthy within the domains of urban planning and urban sociology, where the influence of unobserved variables often presents a significant challenge to predictive modeling. An $R^2$ of this magnitude suggests that over half of the variability in residents' satisfaction can be attributed to the historic built environment features considered in our analysis. This level of explanatory power is substantial, especially given the exploratory nature of our study, which seeks to unravel the nuanced interactions between human satisfaction and environmental factors.

Furthermore, we utilized SHAP (SHapley Additive exPlanations) [65], a method grounded in game theory, to interpret the outputs of our model. SHAP visualizes feature

contributions as forces influencing predictions. Each feature's Shapley value, indicating its impact, is shown as an arrow in Figure 5: red arrows increase and blue arrows decrease the prediction from the baseline average. These forces reach equilibrium at the actual prediction for each data instance. Figure 6 shows a summary of force plots for all test set instances, arranged by similarity and displayed interactively in Jupyter Notebook. This layout helps understand the model's responses to different inputs. The plot's *x*-axis corresponds to data instances, with the size of red and blue areas indicating the magnitude of feature impacts on predictions. Notable clusters, such as around instances 15, 33, and 68, show strong negative, balanced, and positive feature influences, highlighting potential key influencers.

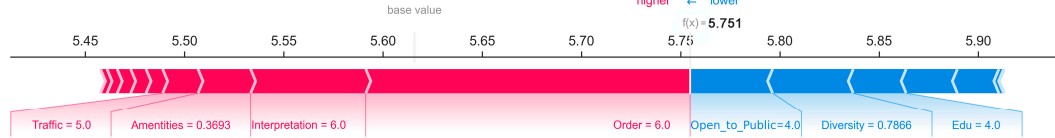

**Figure 5.** An example of the force plot for a single sample instance in the test set.

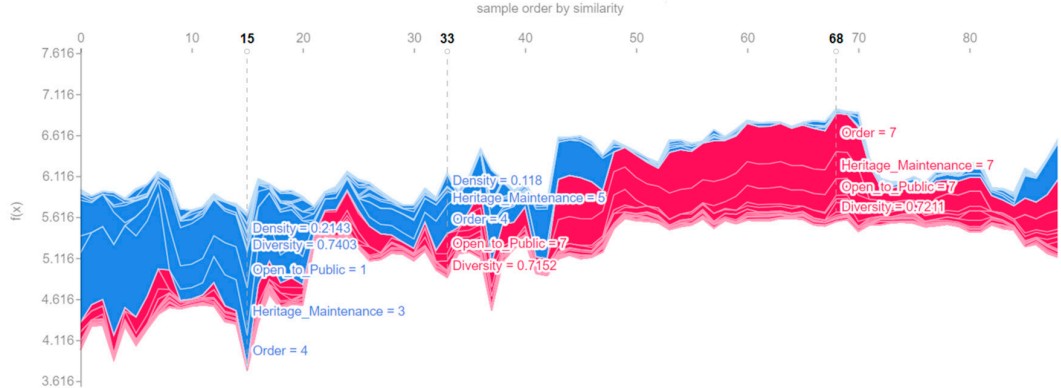

**Figure 6.** The stacked force plot for all samples across the test set.

### 3.2. Relative Contributions of Independent Variables

Figure 7 compares the MDI (Mean Decrease in Impurity)-based feature importance and the permutation importance of all the independent variables. In Figure 7a, the MDI indicates the features that contribute most significantly to the partitioning decisions made within the GBDT model. Order emerges as the predominant feature, suggesting that it plays a critical role in the model's decision-making process. Features such as Density and Income also demonstrate considerable importance, which may imply their strong predictive power within the model. Figure 7b presents permutation importance, which evaluates the impact of feature perturbation on the performance of the model. Unlike MDI, which is intrinsic to the model, permutation importance offers an extrinsic view, examining how the scrambling of feature values affects the accuracy of predictions. Here, Order maintains a position of high importance, although with a wider confidence interval, indicating variability in its impact on model performance. The significance of Interpretation and Openness to Public is notable as well, suggesting that these features' values are crucial to the model's predictive accuracy.

The comparison between MDI and permutation importance highlights a consistent recognition of Order as a key feature, yet the variations in other features' rankings between the two methods suggest differing sensitivities to the features' roles. The differences may arise from permutation importance's ability to capture feature interactions and its robustness to the model's internal structure, which can be particularly insightful for features that may not have high cardinality but interact strongly with other features.

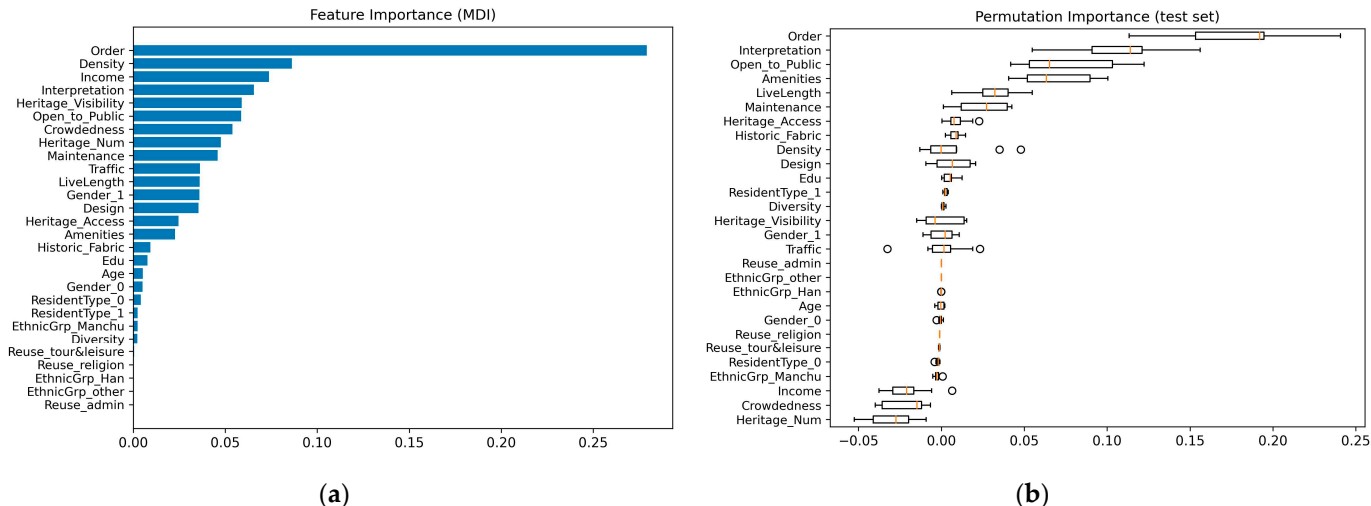

**Figure 7.** Comparative analysis of feature importance. (**a**) MDI based feature importance. (**b**) Permutation importance.

In this study, permutation importance is regarded as a more reliable and preferred method for ranking features. This is attributed to its model-agnostic nature, which tends to provide a more accurate depiction of the practical impact of changes in feature values on the model's predictions. Additionally, it accounts for interactions between features.

In the context of permutation importance, features exhibiting a mean importance greater than zero are identified as key influencers. A positive mean permutation importance implies that the model's performance deteriorates when the feature values are shuffled, indicating that the model relies on these features for making accurate predictions. Consequently, based on permutation importance analysis, 14 features have been identified as key independent variables (see Table 4).

**Table 4.** The relative contributions of the influential variables to residents' satisfaction.

| Feature | MDI Based Importance | Permutation Importance Mean | Permutation Importance Std |
|---|---|---|---|
| Order | 0.279 | 0.173502 | 0.047354 |
| Interpretation | 0.0656 | 0.108591 | 0.02378 |
| Openness to Public | 0.058663 | 0.068155 | 0.033929 |
| Amenities | 0.022648 | 0.064116 | 0.016667 |
| Live Length | 0.036098 | 0.032347 | 0.011704 |
| Density | 0.08628 | 0.024327 | 0.02215 |
| Heritage Maintenance | 0.045824 | 0.020736 | 0.017788 |
| Heritage Accessibility | 0.02459 | 0.008694 | 0.008184 |
| Historic Fabric | 0.00936 | 0.006232 | 0.0037 |
| Design | 0.035429 | 0.005059 | 0.010567 |
| Education | 0.00773 | 0.003857 | 0.005608 |
| ResidentType_1 | 0.002455 | 0.002397 | 0.001023 |
| Diversity | 0.002191 | 0.001264 | 0.000991 |
| Heritage Visibility | 0.058884 | 0.000426 | 0.014148 |

Note: This table includes only variables that have a permutation importance mean greater than zero; 'Resident-Type_1' denotes individuals who were not native-born in Dandong.

Table 4 illustrates the relative contributions of these 14 selected features to residents' satisfaction, based on their positive mean values in permutation importance. Notably, Order emerges as the most significant factor, with the highest Mean Decrease in Impurity (MDI) and permutation importance, underscoring its critical role in the model. Features such as Interpretation, Openness to Public, and Amenities also demonstrate substantial permutation importance means, signifying their considerable influence on the model's

predictions. Interestingly, Live Length and Density, despite lower MDI importance, exhibit meaningful permutation importance, emphasizing their practical impact on predictive accuracy. Conversely, "ResidentType_1", representing non-native-born individuals in Dandong, is still recognized as an influential variable despite its lower importance score, reflecting the intricate role of socio-demographic factors.

### 3.3. Non-Linear Effects of Key Independent Variables

The variability in importance, as evidenced by the confidence intervals in Figure 7b, suggests that certain features may have inconsistent effects across different model instances or data subsets. This insight implies the existence of potential non-linear effects and directs us to focus on the key features for further analysis to enhance model robustness and interpretability.

In this study, partial dependence plots were employed to elucidate the relationships between variables of the historic built environment and residents' satisfaction. PDPs are effective in visualizing the general relationship between the target response and selected features, independent of other variable values. However, PDPs may obscure heterogeneous effects by only showing average marginal effects. To overcome this limitation, PDPs were combined with Individual Conditional Expectation (ICE) plots. ICE plots reveal heterogeneous relationships by illustrating individual prediction paths, thereby highlighting the variability and interactions between features that may be averaged out in PDPs. While ICE plots substantially improve the understanding of model predictions for individual cases, they can become challenging to interpret with a large number of instances, potentially leading to cluttered plots. To address this, we incorporated each PDP along with a shaded area representing the ICE range, as shown in Figure 8. This methodology merges the clarity of PDPs with the detailed insights provided by ICE plots, offering a comprehensive perspective on the model's behavior. In constructing the y-axis of the PDPs for this study, we opted for a representation that emphasizes the relative change in predicted values as opposed to their absolute magnitudes. Specifically, the y-axis in our PDPs reflects the variation in predicted values relative to a baseline, which in this case is the mean prediction across the dataset. This approach allows for a more intuitive understanding of the impact of each feature on the model's predictions, highlighting how deviations from the average prediction are associated with changes in feature values.

In analyzing the relative contributions of independent variables, four perceptually measured discrete variables stand out due to their high feature importance: Order (rank 1), Interpretation (rank 2), Openness to the Public (rank 3), and Heritage Maintenance (rank 6). Figure 8 displays a series of PDPs that elucidate the relationship between these key variables and residents' satisfaction levels. As shown in Figure 8, these PDPs consistently exhibit positive trends across these variables, suggesting a uniform enhancement in resident satisfaction with increased levels of these key features.

Particularly, Interpretation's interaction with resident satisfaction is complex, showing an overall positive trend but initially declining before increasing. Due to Interpretation being a discrete variable, its distribution is not shown on the rug plot. We used the SHAP package for a PDP of Interpretation, with the y-axis showing absolute values and the x-axis including a histogram for sample distribution. Figure 9 reveals low histogram bars in areas of initial PDP decline, indicating fewer samples and suggesting potential misleading results in these sparse data regions. Therefore, the early decline seen in the PDP should be cautiously interpreted, as it might not accurately represent the true relationship between Interpretation and satisfaction due to limited data in these areas.

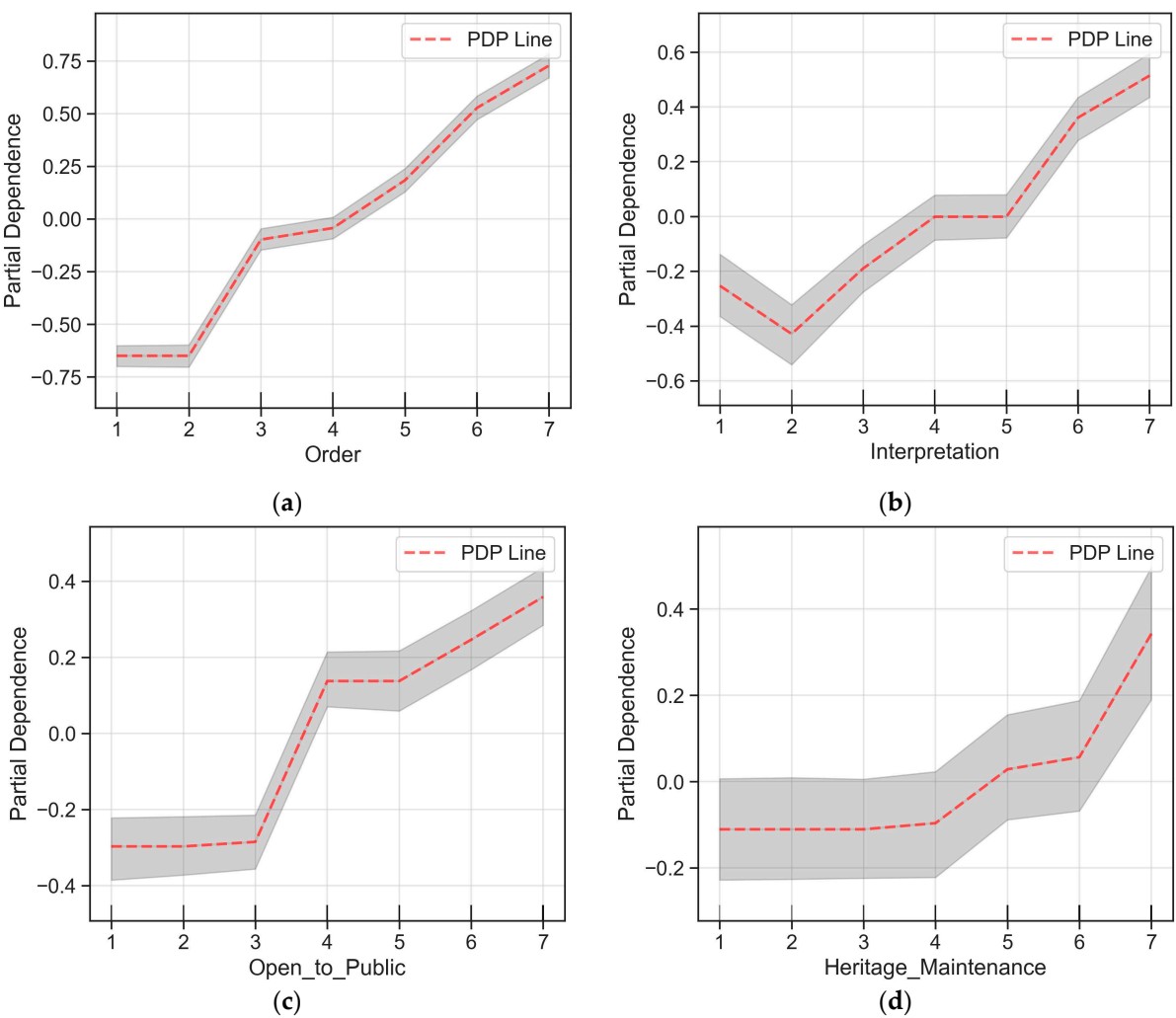

**Figure 8.** Partial dependence plots for perceptually measured discrete variables influencing residents' satisfaction in historic built environments. (**a**) Order; (**b**) Interpretation; (**c**) Openness to the Public; (**d**) Heritage Maintenance.

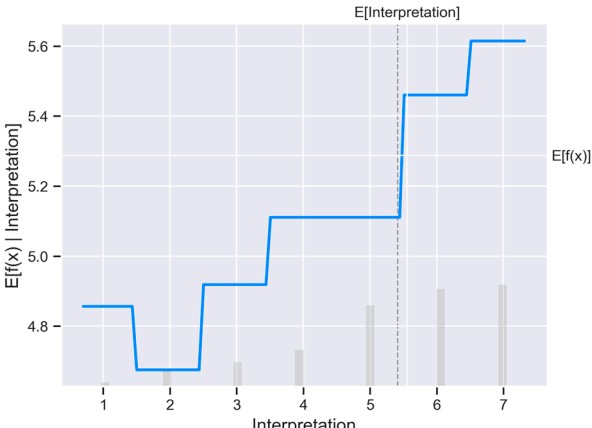

**Figure 9.** SHAP-generated partial dependence plot of Interpretation with an integrated sample histogram.

Figure 10 illustrates the diverse effects of the physically measured characteristics of historic built environments on resident satisfaction. Given that these are continuous variables, a rug plot has been integrated into the visualization. This addition provides a

visual representation of sample density, aiding in the interpretation of the distribution of data points across the range of each feature's values.

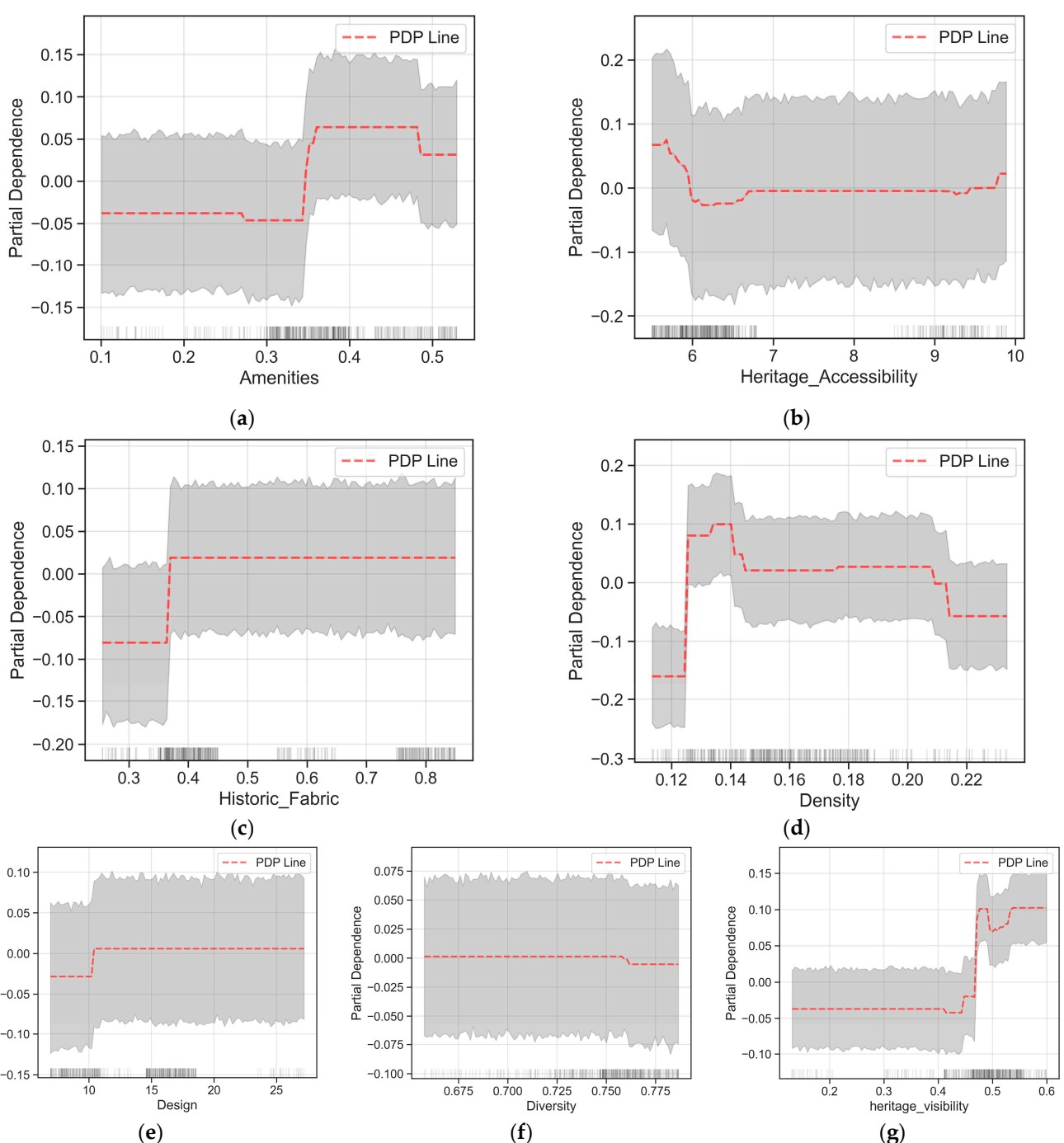

**Figure 10.** Partial dependence plots for physically measured continuous variables influencing residents' satisfaction in historic built environments. (**a**) Amenities; (**b**) Heritage Accessibility; (**c**) Historic Urban Fabric; (**d**) Density; (**e**) Design; (**f**) Diversity; (**g**) Heritage Visibility.

In the plot for Amenities (Figure 10a), we observe a relatively flat trend in the relationship between open space share and resident satisfaction up to the 0.34 mark, suggesting that changes in open space amount have little impact on satisfaction within this range. Between 0.34 and 0.36, there is a significant increase in satisfaction, indicating a key threshold where the effects of open space become more pronounced. Past the 0.36 point, the satisfaction curve flattens again, implying that additional increases in open space do not substantially

boost satisfaction, suggesting a saturation point for optimal open space. The ICE confidence interval around the PDP line shows consistent model predictions across different open space levels.

The Heritage Accessibility curve (Figure 10b) initially shows a decrease in satisfaction with longer travel times to heritage sites. However, after reaching a 6.7 min travel time, further increases do not significantly affect satisfaction. In Figure 10c, the Historic Urban Fabric, measured by the preservation of historical road patterns, shows a notable increase in satisfaction at around 0.368, indicating a strong preference for maintaining historical layouts.

Figure 10d illustrates that satisfaction climbs with building density up to around 0.14, possibly due to advantages like walkability and community feel. Beyond 0.14, satisfaction sharply falls, suggesting resident concerns over issues like overcrowding or loss of open space, until it stabilizes around 0.145. There is another notable drop in satisfaction at around 0.208, leveling off at 0.214, possibly reflecting further perceived declines in living conditions or environmental quality with increased density in a historic context. Beyond this point, further changes in building density do not significantly impact satisfaction.

In the plot for Design (Figure 10e), we see a shift to a positive impact on satisfaction when block density exceeds 10.5 blocks per km$^2$. This may reflect preferences for a certain degree of urban structure that promotes accessibility and street connectivity, contributing positively to the residents' urban experience. As shown in Figure 10f, Diversity's influence on satisfaction remains relatively neutral until it passes the value of 0.758, where it starts to impart a negative effect. This could indicate that, beyond a certain point, too much heterogeneity in the urban fabric may become less appealing to residents.

The plot for Heritage Visibility (Figure 10g) begins with a flat line, indicating that low to moderate levels of Heritage Visibility do not significantly influence resident satisfaction. However, satisfaction sharply increases around the 0.4 mark, continuing until about 0.5, showing a strong positive correlation with heritage visibility. This suggests that a visible connection to heritage significantly enhances residents' perception and enjoyment of their environment. After reaching a peak near 0.5, satisfaction still increases with higher visibility, but at a reduced rate, indicating a potential saturation point where additional visibility has less impact on satisfaction. The plot eventually levels off, showing that extremely high visibility levels do not significantly boost satisfaction further.

Incorporating the nuances of residents' lived experiences within historic built environments, Figure 11 reveals a complex interplay between demographic variables and levels of satisfaction. Length of Residence (Figure 11a) shows stable satisfaction for residents living up to 20 years in the area, but a decline for those residing 21 to 50 years, possibly due to witnessing changes and the loss of familiar features. Educational Attainment (Figure 11b) reveals a trend of decreasing satisfaction up to the college level, then stabilizing at graduate education, indicating that higher education might lead to a more critical view of historic preservation. Figure 11c shows non-native residents have slightly lower satisfaction than natives, suggesting differences in cultural expectations and sense of community. These findings highlight the role of personal, historical, and sociocultural factors in shaping satisfaction in historic urban settings.

### 3.4. Interaction Effects Influencing Non-Linear Relationships

The SHAP dependence plot is employed to further interpret the interactions between the independent variables and their impact on residents' satisfaction. Building upon the nonlinear associations and threshold effects revealed by PDPs, the SHAP dependence plot excels by offering insights into the synergistic dynamics between features. Its inherent capability to automatically detect and illustrate potential interactions is especially advantageous. By color-coding the data points based on an additional feature, the plot elucidates how the influence of one variable may be contingent upon the level or presence of another, effectively capturing the interaction effects. This automatic selection of features for color coding not only enhances the interpretability of complex interactions but also simplifies the discovery of these interdependencies. The SHAP dependence plot thus emerges as

an invaluable tool in our analytical arsenal, facilitating a deeper understanding of the predictive relationships within our model.

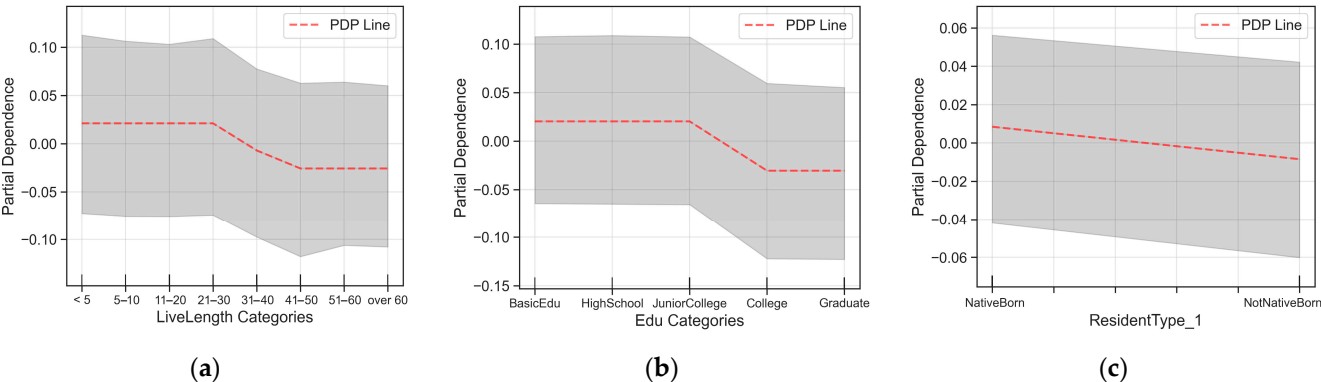

**Figure 11.** Partial dependence plots for key demographic variables influencing residents' satisfaction in historic built environments. (**a**) Live Length (years); (**b**) Education; (**c**) Resident Type.

In the feature importance analysis section, our findings establish Order as the paramount factor affecting residents' satisfaction. The employment of SHAP dependence plots has further accentuated this, revealing clear interaction effects between Order and other variables (Figure 12).

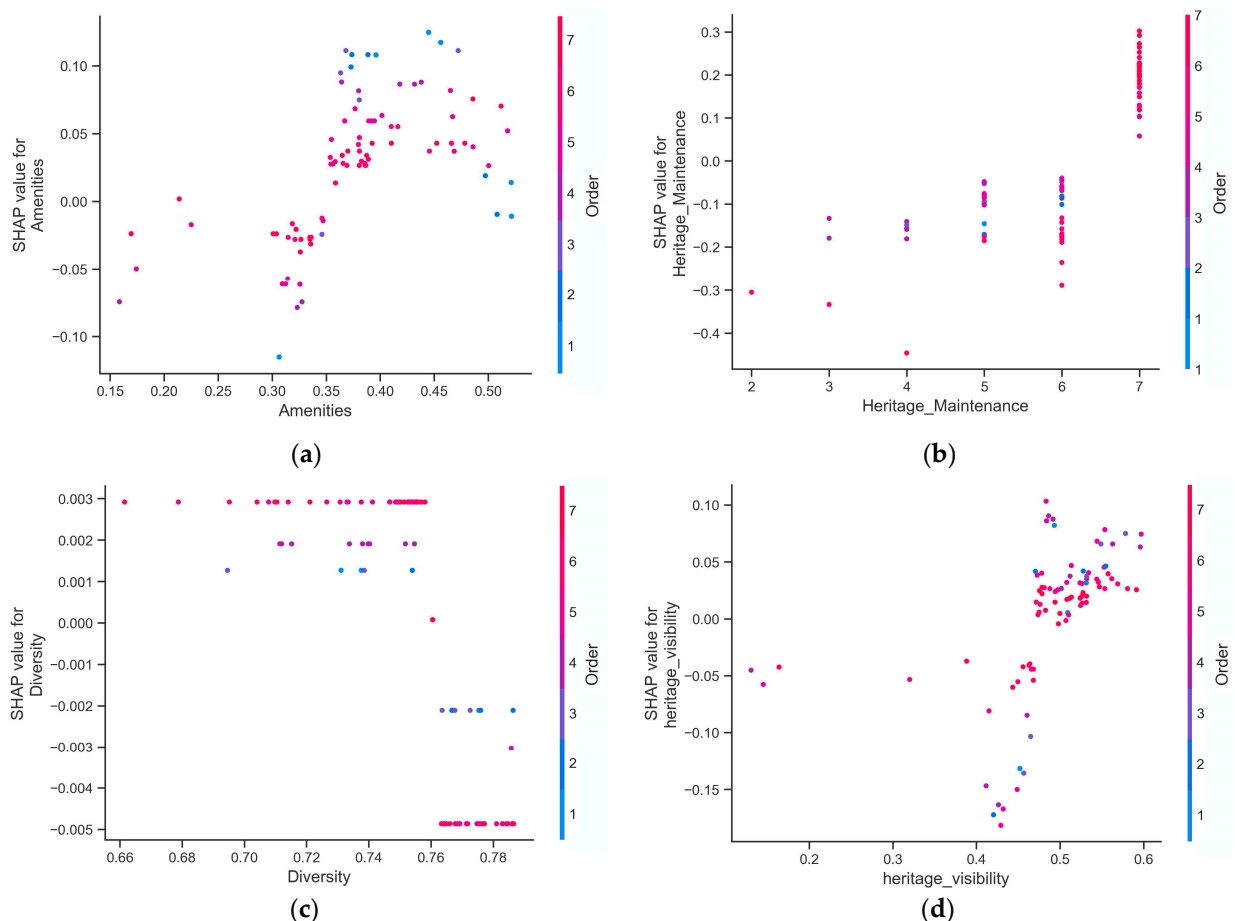

**Figure 12.** SHAP dependence plots demonstrate interaction effects between key variables and Order. (**a**) SHAP dependence plot of Amenities; (**b**) SHAP dependence plot of Heritage Maintenance; (**c**) SHAP dependence plot of Diversity; (**d**) SHAP dependence plot of Heritage Visibility.

The SHAP dependence plot pertaining to Amenities (Figure 12a) demonstrates that, excluding samples with low Order levels, the overall influence of Amenities on satisfaction is on an uptrend. This suggests a contingent relationship where the presence and quality of Amenities significantly bolster satisfaction in an environment where Order is maintained.

For Heritage Maintenance, the dependence plot (Figure 12b) reveals a nuanced interaction. While a high level of Heritage Maintenance correlates with a decent increase in satisfaction, this relationship is heavily modulated by the level of Order. High Order levels augment the positive effects of Heritage Maintenance on satisfaction. Conversely, at lower levels of Order, even optimal Heritage Maintenance does not significantly elevate satisfaction. This phenomenon indicates that the appreciation of maintenance efforts is contingent upon the harmonious and orderly presentation of the environment.

Figure 12c indicates that while higher levels of Order magnify the positive effects of Diversity on residents' satisfaction up to a certain threshold, surpassing it, particularly beyond the 0.758 mark, introduces a negative trend. This suggests that excessive Diversity in the context of high Order may be perceived as chaotic or indicative of a disorganized environment, potentially disrupting the atmosphere in historic settings. The interplay between Order and Diversity highlights the necessity of a balanced approach in urban design, where the benefits of Diversity are supported by sufficient Order to prevent a decline in satisfaction due to perceived disorder.

Heritage Visibility presents a complex interaction with Order, as elucidated in Figure 12d. High Order levels correlate with a clear positive influence on satisfaction as Heritage Visibility increases. However, in low-Order scenarios, the impact is markedly heterogeneous. This could stem from the subjective value placed on heritage sites by different residents. For some, the visibility of heritage evokes positive feelings, particularly when these sites hold personal significance. For others, visible heritage amidst disorder may be perceived as an impediment to progress, leading to negative associations and reduced satisfaction.

Beyond the influence of Order, there are other variables within our study that exhibit significant interaction effects. These interactions contribute to a more comprehensive understanding of the factors shaping residents' satisfaction in historic built environments.

Figure 13a shows that satisfaction related to Heritage Accessibility varies greatly with low public openness, indicating an inconsistent impact on satisfaction when access to heritage sites is limited. As accessibility increases, a clear pattern emerges: satisfaction declines with travel times up to about 6.7 min, then stabilizes. This supports the PDP trend in Figure 10b and may explain variations in that plot, suggesting a threshold beyond which increased accessibility does not notably enhance satisfaction.

Figure 13b reveals that satisfaction linked to Interpretation facilities in historic environments strongly depends on residents' education levels. Higher-educated individuals show more dissatisfaction when Interpretation facilities are deemed insufficient, while well-equipped facilities lead to greater satisfaction among them compared to less educated residents. This implies that more educated residents are more sensitive to the availability and quality of interpretive resources, likely valuing the contextual information they provide at heritage sites.

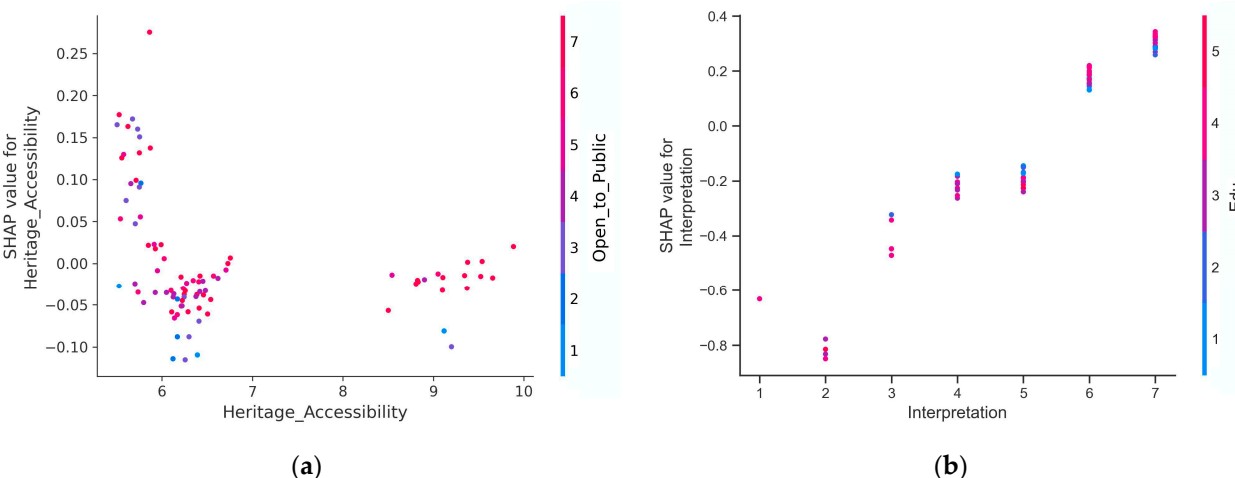

**Figure 13.** Additional SHAP Dependence Plots Highlighting Interaction Effects Among Key Variables. (**a**) Interaction between Heritage Accessibility and Openness to the Public; (**b**) Interaction between Interpretation and Education.

## 4. Discussion

Our study's analysis, demonstrating a significant $R^2$ value of 0.5165, indicates that over half of the variability in residents' satisfaction is associated with the historic built environment features we considered. This result is particularly noteworthy in the fields of urban planning and urban sociology, where the influence of unobserved variables often poses a challenge [66]. The prominence of historic built environment features in our findings highlights their critical role in shaping resident satisfaction. This underscores the need for urban planners and policymakers to consider these elements not just as cultural or aesthetic attributes but as essential factors contributing to the quality of urban life [4,67].

Before further interpreting our findings, it is crucial to acknowledge certain limitations and caveats. Firstly, while our study encompasses characteristics from three main aspects—the built environment's fundamental characteristics; conservation and revitalization; and demographics—it does not include factors from other influential dimensions such as social and economic characteristics. This omission suggests that our analysis might not capture the complete picture of what influences urban residents' satisfaction. Furthermore, this study's focus on a specific historic urban area introduces a limitation regarding the generalizability of our findings. Each city has its own unique environmental features and cultural differences, meaning the associations and thresholds identified in our study may vary in different urban contexts. Therefore, caution should be exercised in applying these findings to other cities without considering their specific characteristics. Despite these limitations, our study offers valuable insights into how the historic built environment influences residents' satisfaction. It provides a fresh perspective that can inform the management and development of historic cities, emphasizing the need to consider a wide range of factors, including those pertaining to heritage and cultural significance. Furthermore, our study concentrated solely on exploring the impact of the physical attributes of the historic built environment on residents' satisfaction. Nevertheless, it is essential to acknowledge that other research underscores the crucial influence of immaterial aspects. These include cultural or historical significance, a sense of place and belonging, and the extent of identification with the built environment, all of which play a pivotal role in shaping satisfaction and overall well-being [44,68].

Our study encountered challenges in measuring key aspects of the historic environment like Heritage Maintenance, Openness to the Public, Interpretation, Order, and nuisances related to Traffic and Crowdedness. The difficulty was in physically quantifying these subjective elements. We used perception data from questionnaires as a proxy for these physical conditions. Remarkably, four perceptually measured variables—Order, Interpretation, Openness to the Public, and Heritage Maintenance—were highly signif-

icant in our analysis; likely because they reflect residents' direct experiences with their environment [69].

We combined objective physical measurements with subjective perception data to understand their impact on resident satisfaction. However, some theories suggest a sequential process where individuals first observe objective features, which are then interpreted subjectively, influencing satisfaction [19,20]. This perspective implies a complex interaction where objective reality is filtered through personal views before affecting satisfaction. Our study did not explore this sequential dynamic, pointing to a future research opportunity using structural equation models to examine how physical environments and individual perceptions interact in shaping satisfaction in historic urban contexts.

Order emerged not only as the most influential variable but also as one with significant interaction effects with other key variables. The importance of Order is primarily rooted in its comprehensive nature, encapsulating a range of elements crucial to the historic built environment. These include aesthetics, contextual harmony, local identity, continuity of local characteristics, and the balance between the old and new. Such a broad scope makes Order a representation of the overall atmosphere and character of a historic urban area. Research in environmental psychology suggests that environments that blend harmoniously with their cultural context and possess aesthetic appeal enhance subjective well-being, alleviate stress, and promote improved social interactions [49,70]. Maintaining Order in the historic built environment is therefore of paramount importance. It goes beyond mere organization or tidiness; it is about preserving the continuity and harmony of the area's historical context while accommodating modern needs [71,72]. This maintenance ensures that the intrinsic value of the historic environment is upheld, contributing significantly to the residents' sense of belonging, identity, and satisfaction. The interplay of Order with other factors further accentuates its role, indicating that its maintenance can have a ripple effect on various aspects of the urban living experience.

Our analysis of the non-linear effects of key independent variables in historic built environments uncovers several pivotal findings, each carrying distinct implications for urban planning. Firstly, the relationship between heritage accessibility and resident satisfaction, which stabilizes after a 6.7 min travel time, underscores the importance of proximity in heritage conservation but also indicates a limit to the value placed on accessibility. The appreciation for the preservation of historical road patterns, becoming notably significant at a value of around 0.368, suggests residents' preference for maintaining authentic historical elements in urban planning. Moreover, this study reveals a nuanced preference for urban compactness, with satisfaction increasing up to a building density of 0.14 and then declining, indicating the need for a balanced approach in urban density planning. This is further reinforced by the shift in satisfaction when block density exceeds 10.5 blocks per km$^2$, suggesting a preferred degree of urban structure that balances accessibility, connectivity, and historic character preservation. Interestingly, the satisfaction levels remain neutral until they surpass a diversity value of 0.758, after which they negatively impact satisfaction, pointing to the complex balance needed to integrate diversity into the urban fabric. Excessive heterogeneity may be perceived as disruptive to the historic character and resident satisfaction. The most striking finding is the sharp increase in satisfaction at a Heritage Visibility level of approximately 0.4, continuing until about 0.5. This underscores the significant impact of visual connections with heritage sites on resident satisfaction and highlights the value of not only preserving but also ensuring the visibility of heritage sites in the urban landscape. This aspect is crucial for enhancing cultural engagement and aesthetic appreciation among residents.

Although architectural and design elements significantly impact personal satisfaction in urban environments, an individual's cultural background also significantly contributes to their environmental satisfaction [73]. The decrease trend in Live Length might reflect the impact of witnessing changes over time, where prolonged familiarity and deep connections with the environment diminish due to alterations in the historic fabric and the loss of familiar, memory-laden features. The association pattern of Education indicates a decline

in satisfaction from the middle school level up to the college level. This trend may reflect differing expectations based on educational attainment, where higher education levels could correspond with a more critical view of the historic built environment or a greater desire for modern amenities. While the satisfaction of non-native residents is marginally lower compared to native residents, suggesting that non-native residents tend to have a lower satisfaction level with the historic built environment compared to native residents. This could be due to various factors, such as cultural differences, a sense of belonging, or differing values and expectations regarding historic significance and its preservation. However, it is important to note that this study primarily focuses on architectural and design elements, and as such, the influence of personal characteristics on satisfaction in historic urban environments warrants further detailed investigation in future research.

In our study, we employed PDPs along with ICE ranges to analyze the data. This combination is crucial for overcoming some of the inherent weaknesses of PDPs. PDPs alone can sometimes provide a somewhat limited view, as they average out the effects of the features across all samples, potentially obscuring individual variations. The inclusion of ICE ranges alongside PDPs allows us to capture individual variations in the data, offering a more nuanced understanding of how different factors influence residents' satisfaction. However, it is important to note that even this improved methodology has its limitations. For instance, PDPs might consider ranges of the target variable that are not realistic, such as negative building density. This issue highlights the potential need for future studies to incorporate Accumulated Local Effects (ALE) plots [74]. ALE plots can address this limitation by focusing on the actual range of data, providing an even more accurate and realistic understanding of the effects of various features on residents' satisfaction.

Our study identified various interaction effects among the variables, which suggests a complex interplay in how these factors collectively influence residents' satisfaction. However, one aspect that remains unclear is the existence and nature of potential synergy effects among these variables. Synergy effects occur when the combined impact of two or more variables is greater than the sum of their individual effects. Understanding whether and how these synergy effects exist in the context of the historic built environment could provide deeper insights into optimizing urban planning and policy decisions [63,75]. For instance, if certain combinations of features like Heritage Maintenance and Openness to the Public have a synergistic effect on satisfaction, urban planners could prioritize these combinations in their development strategies. Future research should explore this area, potentially employing more complex analytical models to unravel these synergistic relationships. Such studies would not only contribute to the academic discourse in urban planning and sociology but also provide practical guidance for enhancing the quality of life in historic urban areas.

## 5. Conclusions

This research, utilizing data from the old town of Dandong, China, explores the factors influencing residents' satisfaction with historic built environments and identifies the effective ranges in which key factors correlate with satisfaction. The Gradient Boosting Decision Trees technique is employed to assess variables related to satisfaction derived from historic built environment features and demographic factors. The GBDT method excels in revealing the varying influences of different factors and their complex, non-linear relationships with residents' satisfaction while concurrently controlling for other variables. The findings from this study offer valuable contributions to the existing literature, deepening our understanding of the nuanced interactions between residents and their historic built environments.

The variables examined in this study are instrumental in predicting residents' satisfaction, with the combined contribution of historic built environment features and demographic factors totaling 51.65%. Although demographic factors are important, their influence is markedly lower compared to that of historic built-environment features. This

suggests that a significant portion of the variability in residents' satisfaction is predominantly attributed to the historic built environment features analyzed.

Among the perceptually measured variables in this study, four exhibit particularly high feature importance: Order, Interpretation, Openness to the Public, and Heritage Maintenance. The relationship between these key variables and residents' satisfaction levels consistently displays positive trends. This indicates a uniform increase in resident satisfaction correlating with heightened levels of these key features. Notably, Order not only emerged as the most influential variable but also demonstrated significant interaction effects with other key variables. The interaction of Order with other factors further underscores its pivotal role, suggesting that its effective management can profoundly influence various facets of the urban living experience.

In this study's analysis of physically measured characteristics, we observed pronounced non-linear effects in their association with residents' satisfaction. Additionally, several insightful thresholds have been identified, enhancing our understanding of the impact of these characteristics on satisfaction levels. For instance, the share of open space significantly boosts satisfaction when it exceeds 34%. Travel time to heritage sites reveals a decreasing trend in satisfaction with increases, stabilizing after approximately 6.7 min. This suggests that beyond this duration, further increases in travel time do not notably alter satisfaction levels.

Regarding historic urban fabric, a significant positive impact on satisfaction becomes apparent at a value of around 36.8%, indicating a clear preference for the preservation of historical road patterns. Building density positively influences satisfaction up to about 14%, likely owing to benefits such as improved walkability and a stronger community feel. However, satisfaction sharply decreases as density surpasses this value until it plateaus around 14.5%, reflecting discomfort with higher density in a historic context. A positive shift in satisfaction is observed when block density exceeds 10.5 blocks per $km^2$, likely due to preferences for a degree of urban structure that enhances accessibility and street connectivity. The satisfaction influence remains neutral until the entropy index for diversity surpasses 0.758, after which it exerts a negative effect. This implies that excessive heterogeneity in the urban fabric may become less appealing beyond this threshold.

Moreover, low to moderate levels of heritage visibility do not significantly impact satisfaction. However, a marked increase in satisfaction is observed when the proportion of areas from which heritage can be seen reaches approximately 40%, continuing until about 50%. This pattern suggests that beyond a certain threshold, the visual connection with heritage sites becomes a significant contributor to residents' satisfaction, potentially due to increased cultural engagement or aesthetic appreciation.

This study highlights key learnings for urban planners and decision-makers in historic environments. It emphasizes the importance of improving historic built environments to enhance residents' satisfaction. Prioritizing the key attributes identified, such as Order and Openness to the Public, is crucial for managing historic cities effectively. Additionally, the identified thresholds, like maintaining over 34% open space and ensuring block density above 10.5 blocks per $km^2$, offer valuable benchmarks for developing quantitative design codes and management policies in areas like Dandong's old town.

**Supplementary Materials:** A comprehensive explanation of the core principles and the detailed algorithm of GBDT is available online. For an in-depth understanding, please refer to Video S1: https://www.youtube.com/watch?v=3CC4N4z3GJc (11 December 2023); Video S2: https://www.youtube.com/watch?v=2xudPOBz-vs (11 December 2023); Document S3: https://scikit-learn.org/stable/modules/generated/sklearn.ensemble.GradientBoostingRegressor.html (11 December 2023).

**Author Contributions:** Conceptualization, X.J.; methodology, X.J.; software, X.J. and Y.D.; validation, X.J. and Q.L.; formal analysis, X.J., Y.D. and Q.L.; investigation, X.J. and Y.D.; resources, X.J., Y.D. and Q.L.; data curation, Y.D.; writing—original draft preparation, X.J.; writing—review and editing, Y.D. and Q.L.; visualization, X.J. and Y.D.; project administration, X.J.; funding acquisition, X.J. All authors have read and agreed to the published version of the manuscript.

**Funding:** This research was funded by the National Natural Science Foundation of China (No. 52208046) and the Fundamental Research Funds for the Central Universities (No. N2211003 and No. N2211002).

**Institutional Review Board Statement:** Not applicable.

**Informed Consent Statement:** Not applicable.

**Data Availability Statement:** The data presented in this study are available from the authors upon reasonable request.

**Acknowledgments:** The authors would like to thank Jason Cao of the University of Minnesota for his enlightenment on the original ideas of the research. The authors appreciate the anonymous reviewers for their valuable comments and suggestions.

**Conflicts of Interest:** The authors declare no conflict of interest.

## Appendix A. The Algorithm of Gradient Boosting Decision Trees

The description of the algorithm was borrowed heavily from previous research [18,60,76]. The Gradient Boosting Decision Trees (GBDT) algorithm constructs a robust predictive model by iteratively adding decision trees. Each subsequent tree in the sequence aims to correct the residual errors of its predecessors, guided by the gradient of a designated loss function. The GBDT algorithm's process for regression tasks can be summarized as follows:

Input: Data $\{(x_i, y_i)\}_{i=1}^n$, and a differentiable Loss Function $L(y_i, F(x))$.

Step 1: Initialize the model with a constant value:

$$F_0(x) = \underset{\gamma}{argmin} \sum_{i=1}^n L(y_i, \gamma), \tag{A1}$$

where $y_i$ represents the observed values and $\gamma$ represents the predicted values.

Step 2: form m = 1 to M (m refers to the number of an individual tree):

(A) Compute:

$$r_{im} = -\left[\left(\frac{\partial L(y_i, F(x_i))}{\partial F(x_i)}\right)\right]_{F(x)=F_{m-1}(x)} \text{ for i = 1, 2, …, n,} \tag{A2}$$

where $r_{im}$ represents the pseudo residual.

(B) Fit a regression tree to the $r_{im}$ values and create terminal regions $R_{jm}$, for j = 1, 2, …, $J_m$.

(C) For j = 1, 2, …, $J_m$, compute:

$$\gamma_{jm} = \underset{\gamma}{argmin} \sum_{x_i \in R_{ij}} L(y_i, F_{m-1}(x_i) + \gamma). \tag{A3}$$

(D) Update:

$$F_m(x) = F_{m-1}(x) + \vartheta \sum_{j=1}^{J_m} \gamma_{jm} I(x \in R_{jm}), \tag{A4}$$

where $\vartheta$ refers to the learning rate.

Step 3: Output $F_M(x)$.

## Appendix B. Survey Questionnaire on Satisfaction with the Historic Built Environment of the Old Town of Dandong

Note: The following information is translated from the original questionnaire, which was in Chinese.

1.  Which of the following best describes your residential status in the Dandong Old Town area? [Single Choice]

    ☐ Born and raised (Born in Dandong and have long-term residence in the Old City area)
    ☐ Settled from elsewhere (Born in another city and later moved to Dandong for long-term residence)

  ☐ Visitor (Having previously lived in the Dandong Old City area for a long time but currently settling elsewhere, occasionally returning home)

2. Please select the length of your residence in the Dandong Old Town area: [Single Choice]

  ☐ Less than 5 years
  ☐ 5–10 years
  ☐ 11–20 years
  ☐ 21–30 years
  ☐ Over 30 years

3. Please enter the name or address of your residential community:
4. Please enter the name or address of your workplace:
5. Please enter your usual shopping location(s) or address(es):
6. Please enter your usual locations for leisure, walking, or entertainment:
7. Do you have any other frequent destinations? If yes, please enter their name(s) or address(es):
8. How strongly do you agree or disagree with the following statements about the historic urban environment that forms the backdrop of your everyday life? [Matrix Scale Question]

| Statements | 1 Extremely Not True | 2 Quite Not True | 3 Slightly Not True | 4 Not Sure | 5 Slightly True | 6 Quite True | 7 Entirely True |
|---|---|---|---|---|---|---|---|
| The old buildings (or old bridges and other structures) in this historic area are well-maintained. | ☐ | ☐ | ☐ | ☐ | ☐ | ☐ | ☐ |
| The interpretive facilities (e.g., panels, posters, inscriptions) explaining the historical background in this historic area are comprehensive. | ☐ | ☐ | ☐ | ☐ | ☐ | ☐ | ☐ |
| The surrounding buildings and other elements of this historic area have a good harmony with the overall historical atmosphere. | ☐ | ☐ | ☐ | ☐ | ☐ | ☐ | ☐ |
| There is a high volume of vehicular traffic in the historic area, along with issues of traffic noise. | ☐ | ☐ | ☐ | ☐ | ☐ | ☐ | ☐ |
| The historic area experiences a high volume of pedestrian traffic, resulting in a crowded and noisy environment. | ☐ | ☐ | ☐ | ☐ | ☐ | ☐ | ☐ |

9. How would you rate the accessibility and openness of the buildings in this historic area to the public? [Matrix Scale Question]

| | 1 Not Open | 2 High Fee | 3 Moderately High Fee | 4 Reasonable Fee | 5 Low Fee | 6 Very Low Fee | 7 Free Access |
|---|---|---|---|---|---|---|---|
| (Choose the level of openness based on the descriptions) | ☐ | ☐ | ☐ | ☐ | ☐ | ☐ | ☐ |

10. Overall, how satisfied are you with this historic urban environment you live in? [Matrix Scale Question]

| | 1 Strongly Dissatisfied | 2 Dissatisfied | 3 Slightly Dissatisfied | 4 Neutral | 5 Slightly Satisfied | 6 Satisfied | 7 Strongly Satisfied |
|---|---|---|---|---|---|---|---|
| (Higher numbers indicate higher satisfaction) | ☐ | ☐ | ☐ | ☐ | ☐ | ☐ | ☐ |

11. Your Gender: [Single Choice]

  ☐ Male
  ☐ Female

12. Please select your ethnicity: [Single Choice]

☐ Han
☐ Manchu
☐ Korean
☐ Hui
☐ Other ethnicities

13. Your Age Group: [Single Choice]

☐ Under 18
☐ 18–25
☐ 26–35
☐ 36–40
☐ 41–50
☐ 51–60
☐ Over 60

14. Your Educational Level: [Single Choice] *

☐ Middle school and below
☐ High school (or equivalent)
☐ Junior college (or equivalent)
☐ Bachelor's degree (or equivalent)
☐ Graduate degree and above

15. Your Household Monthly Income Range (including all family members living together and any allowances received from other family members): [Single Choice] *

☐ 2000 RMB and below
☐ 2001–4000 RMB
☐ 4001–6000 RMB
☐ 6001–10000 RMB
☐ Over 10,000 RMB

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
