# Peer review of "How Does the Historic Built Environment Influence Residents’ Satisfaction? Using Gradient Boosting Decision Trees to Identify Critical Factors and the Threshold Effects"

_sustainability, doi:10.3390/su16010120_

Round 1
Reviewer 1 Report
Comments and Suggestions for Authors
I have read the article entitled as "How Does the Historic Built Environment Influence Residents’ Satisfaction? Using Gradient Boosting Decision Trees to Identify Critical Factors and the Threshold Effects"
It is a good paper with important merits. However, I suggest improving the paper in the following directions.
Method section should be augmented with equational presentations.
Engagement with recent litetature is not satisfactory. Extend the literature review.
Provide discussion with recent studies and possible policy implications.
Robustness should be provided or discussed.
Also, include limitations and future directions.
Comments on the Quality of English Language
Minor typos and grammar issues.
Reviewer 2 Report
Comments and Suggestions for Authors
Dear authors, thank you for submitting the manuscript to the journal Sustainability.
I understand that in this paper authors use Boosting Decision Trees to identify factors affecting residents' satisfaction in historic built environments.
The topic is very interesting and may have relevant policy implications, both at the national and international level. I consider the paper connected with the overall philosophy of the Journal.
The authors have clearly done a lot of work, all terms were coherently explained, they have set out some interesting hypotheses, leading to some tentative conclusions. In what follows, I shall try to point out my main concerns.
-please explain other methods that can be used, besides Gradient Boosting Decision Trees
- In my opinion, the literature review should be updated, many references are out of date (1976,1983,1994…)
Good luck.
Reviewer 3 Report
Comments and Suggestions for Authors
See attached

Reviewer 4 Report
Comments and Suggestions for Authors
This paper is well written, gradient boosting decision trees is used to identify the critical factors and thresholds of historical buildings' satisfaction with residents. This research topic is innovative, it focuses on the relationship between the historical built environment and residents' satisfaction, and fully reflects the people-oriented thought. The nonlinear relationship between them has high research value and is also an important research content of this paper.
There are some problems, which must be solved before it is considered for publication. If the following problems are well-addressed, this reviewer believes that the essential contribution of this paper are important for urban environment.
The biggest problem with this article is that it is too long. Although the article is detailed in describing variables and graphs, the text is a bit lengthy. In order to better convey the essence of the research, it is recommended to compress the space as much as possible and keep only the key core content. For example, in the description of the analysis method in section 2.3, some unnecessary details can be omitted and only the innovative and essential parts can be emphasized. In this way, readers can understand the research methods and conclusions more quickly and improve the reading experience.
When describing the innovative contribution, the author is too conservative and does not fully highlight its importance and uniqueness. The author makes a comprehensive review of the existing researches and puts forward a new theoretical framework and viewpoints on this basis. However, when the authors introduce these theoretical frameworks and viewpoints, they often simply mention their differences from existing research, without in-depth explanation of their innovation and importance. This makes it difficult for readers to understand the actual value of the paper and to distinguish it from other similar studies.
In addition, the quality of some pictures in the paper is not clear enough. This may be because the image is over-compressed, the image has been improperly processed, or the image is exported at a low resolution. In many images, there is some blurring around the text, and there are shadows around the dots in the scatter plot, which may affect the reader's assessment of the accuracy and credibility of your research results. Therefore, I recommend that you consider replacing these images and make sure to use a high quality image format and proper image processing to make them clearer, easier to read and understand.
Reviewer 5 Report
Comments and Suggestions for Authors
The primary objective of this academic manuscript is to discern critical factors and threshold effects influencing residents' satisfaction in historic cities. The research employs Gradient Boosting Decision Trees, a machine learning technique adept at unveiling nonlinear relationships and interactions among variables. The study yields a comprehensive set of fifteen attributes across four dimensions, chosen for further investigation, along with a predictive model for residents' satisfaction derived from these attributes. The study's conclusions highlight the integral role of residents' satisfaction in their overall well-being within historic environments. It emphasizes that understanding and prioritizing residents' satisfaction in the historic built environment extend beyond managing physical spaces, contributing significantly to enhancing the overall quality of life for urban residents. Notably, the research identifies specific features in the historic environment, such as accessibility and cultural significance, exerting a more pronounced impact on residents' satisfaction than others. The study advocates for prioritizing local residents' satisfaction in urban conservation efforts and involving them in decision-making processes. While the manuscript boasts a well-structured format and robust methodologies, a critical assessment reveals that sophisticated methods were employed to address a seemingly trivial problem, diminishing its scientific significance for publication in a high-quality journal. Given the financial support from China's National Natural Science Foundation, renowned for funding studies addressing "scientific problems", a crucial inquiry arises: What scientific problem does the study aim to resolve, and is this problem of sufficient scientific value? These considerations underscore the need for a clearer delineation of the study's contributions to the academic landscape. Furthermore, for the sake of transparency and replicability, it is advisable to include the questionnaire used for the survey as an appendix. This addition would allow readers and researchers to gain a comprehensive understanding of the survey design and facilitate potential replications of the study. Addressing these concerns will strengthen the manuscript's overall credibility and relevance in the academic domain
Comments on the Quality of English LanguageThe language used in the manuscript does appear somewhat unnatural in English, resembling a translation from Chinese. The authors may need to enhance naturalness of the language.
Round 2
Reviewer 1 Report
Comments and Suggestions for Authors
The revisions are well made and after the changes, the paper is improved. My decision is positive for this last version after revisions. Congrats to the authors.
Reviewer 4 Report
Comments and Suggestions for Authors
All the comments are worked out by the authors. And this edition of manuscript is appropriate for publishing.
Reviewer 5 Report
Comments and Suggestions for Authors
All my comments have been addressed. Thus I recommed acceptance.